# Influence of Soil Physicochemical Properties and Inter-Root Microbial Communities on the Inhibition of Anthracnose in Peppers

**DOI:** 10.3390/microorganisms13030661

**Published:** 2025-03-14

**Authors:** Yongbin Ma, Miaomiao Liu, Yuting Hong, Yichao Wang, Xiaoke Chang, Gongyao Shi, Huaijuan Xiao, Qiuju Yao, Fan Yang

**Affiliations:** 1Horticulture College, Henan Agricultural University, Zhengzhou 450001, China; nky202501@163.com (Y.M.); 15938225659@163.com (Y.W.); xhj234@126.com (H.X.); 2Agricultural College, Zhengzhou University, Zhengzhou 450001, China; liumm2023@163.com (M.L.); hongyt182808@163.com (Y.H.); shigy@zzu.edu.cn (G.S.); 3Institute of Vegetables, Henan Academy of Agricultural Sciences, Graduate T&R Base of Zhengzhou University, Zhengzhou 450002, China; cxk8802@163.com

**Keywords:** pepper, anthracnose, soil properties, high-throughput sequencing, rhizosphere microbial communities

## Abstract

Anthracnose is a widespread plant disease affecting vegetables, flowers, crops, and fruit trees, causing significant economic losses. It occurs at various stages of pepper growth, leading to rotting and shedding in later stages. The aim of this study was to explore the relationship with anthracnose occurrence by analyzing the physicochemical properties and microbiota changes in the inter-root soil of pepper under different susceptibility levels to reveal the key microecological factors and dominant microbial populations and to provide reference for ecological control. Illumina Miseq sequencing was first used to evaluate the physicochemical properties and microbial taxa in pepper inter-root soil across different health statuses and identify key parameters associated with anthracnose. Subsequently, PICRUSt2 (systematic genetic Investigation of communities by Reconstruction of observed States 2) and FUNGuild (Fungi Functional Guild) V1.0 online platform were used to predict the activities of inter-root bacteria and fungi. The findings indicated that healthy peppers had significantly higher inter-root soil nutrient levels and enzyme activity compared to sensitive peppers. There were significant differences between their community structures. In alpha-diversity analysis, inter-root soil microbial richness and diversity were significantly higher in healthy peppers than in susceptible peppers. At the bacterial taxonomic level, the comparative prevalence of Acidobacteria in highly resistant plants, resistant plants, and susceptible plants decreased sequentially. At the genus level, the relative abundance of *Vicinamibacteraceae* and *RB41* was markedly elevated in disease-resistant inter-root soils than in disease-susceptible soils. At the fungal level, the comparative prevalence of *Ascomycetes* in highly resistant plants, resistant plants, and susceptible plants increased sequentially. Differences in function are mainly manifested in apoptosis and mycelial development.

## 1. Introduction

*Capsicum annuum* (*Capsicum annuum* L.) is commonly known as the chili pepper or sweet pepper. *Capsicum* is rich in beneficial substances such as capsaicin, lutein, and β-carotene and is a cash crop widely grown worldwide [1]. At the same time, pepper is also one of the most widely planted vegetables in China, planted all over the country, and can be produced annually [2]. Anthracnose is a prevalent plant disease caused by *Colletotrichum* spp., which can seriously harm a variety of crops, vegetables, flowers, and fruit trees, causing significant economic losses [3]. When the disease occurs, yields are usually reduced by 30–50 percent, and severely affected fields can even result in crop failure [4]. The occurrence of pepper anthracnose has the characteristics of wide distribution, heavy damage, and high spreading speed [5]. It occurs both at planting and post-harvest, mainly affecting pepper fruits but sometimes also on stalks and leaves, with the disease appearing as brown or dark-brown rounded sunken spots. The latter stages of the disease usually cause rotting and shedding of the entire pepper [6]. The main pathogens that cause anthrax in peppers are Bacillus glabrata, Bacillus niger, Bacillus nigra, and Bacillus nigricans [7]. These pathogens are able to endure in the soil for extended periods and have the capacity to withstand and acclimate to adverse situations [8]. The disease has become an important constraint on modern agricultural development. Although chemical control is a fast and efficient means of pest control, it has many problems. Excessive use of chemical pesticides can seriously interfere with the inter-root microbial community, making the ecological environment more fragile. This not only leads to the loss of agrobiodiversity but also causes environmental pollution, which in turn jeopardizes human and animal health. In addition, the long-term use of chemical pesticides may also lead to increased pathogen resistance, excessive pesticide residues, and the re-emergence of pests. Consequently, the quest for innovative and efficient biopesticides to supplant chemical pesticides has emerged as a prominent focus of contemporary research.

The inter-root environment is the primary zone for information and material exchange among microorganisms, plants, and soil [9]. Inter-root microorganisms significantly impact biochemical cycle processes in the soil and are essential for root health and development [10]. Plants can recruit beneficial and pathogen-antagonistic microorganisms through root secretions or the autoimmune system, thereby remodeling the inter-root microbial community, increasing their resistance to pathogens and protecting themselves from infection [11,12]. Inter-root microorganisms affect plants most closely [13]. The inhibitory effects of these inter-root microorganisms on pathogens have been reported several times, and the patterns include hyperparasitism [14], the release of antibacterial substances [15], and competitive competition for resources, including nutrients or space, to suppress harmful microbes [16]. In the past few years, researchers have identified several bacteria from plant tissues or soil that can inhibit different plant pathogens. For example, *Actinomycetes* can significantly inhibit the growth of hyphae, thereby suppressing bacterial wilt [16]. A highly significant positive correlation has been shown between the occurrence of bitter melon wilt and the number of *Fusarium acanthamoeba* in the soil [17]. In addition, compounding different strains to form a microflora can better inhibit pathogenic microorganisms. The multi-strain synthetic flora was more effective in the control of blight, the disease index of wilt was significantly reduced, and plant growth and development were promoted [18]. An endophytic bacterial community of *Dendrobium ferrugineum* improves resistance to soft rot disease in ginger [19]. Based on previous studies, it was found that healthy samples had more diverse microbiological communities than diseased samples [20]. This is because inter-root microbial interactions form a complex microbial ecological network structure that inhibits pathogenic bacteria and creates a healthier soil microbiological environment [21]. Therefore, studying the composition and structure of inter-root microbial communities is important to assess their ability to resist pathogen invasion. Nonetheless, it remains ambiguous if disparities exist in the inter-root bacterial communities between disease-resistant and disease-susceptible species of *Capsicum* and whether these differences correlate with the extent of *Capsicum* anthracnose incidence.

It is unclear how specific microorganisms in the inter-root soil of pepper affect plant health. Accordingly, the focus of this experiment was to detect anthracnose-resistant soil in pepper and to investigate the relationship between their disease-suppressive properties and inter-root soil microbes. The aims of the present study were as follows: (i) to analyze the changes of environmental factors and microbiota in the inter-root soil of pepper in three different health states; (ii) to predict the interactions among anthracnose, soil factors, and inter-root microbial communities. Our research hypothesis was to assess the potential of microecological precision modulation as a basis for preventing anthracnose in the future.

## 2. Materials and Methods

### 2.1. Sample Collection

The pepper variety used in this experiment was “Sanying-9 pepper”, which was planted in the experimental base of the Henan Academy of Agricultural Sciences for producing and growing peppers. To investigate how anthracnose affects the peppers’ inter-root microbial community, we sampled soils with different levels of incidence. The designated groups were as follows: HR (highly resistant plants), R (resistant plants), and S (susceptible plants). Areas with 0% incidence are highly resistant, 0–5% incidence is resistant, and more than 50% incidence is susceptible. The inter-root soil was collected using the S-type five-point random sampling method [22]. That is, five soil samples were collected and combined into a composite sample using an S-shaped sampling track. For the purpose of extracting DNA and characterizing the soil, all samples of soil were rapidly frozen in liquid nitrogen for an hour before being kept in a refrigerator set at −80 °C.

### 2.2. Determining the Chemical Characteristics and Enzyme Activity of Soil

A total of 3 technical replicates were conducted to evaluate the 16 physicochemical characteristics of HR, R, and S inter-root soils in order to evaluate the soil conditions in the study region. These qualities included quick-acting nitrogen (QN), quick-acting phosphorus (QP), quick-acting potassium (QK), calcium/magnesium content (Ca/Mg), zinc content (Zn), iron content (Fe), copper content (Cu), soil pH, organic matter (OM), soil electrical conductivity (EC), microbial carbon biomass (MBC), soil sucrase (S-SC), soil alkaline phosphatase (S-ALP), soil urease (S-UE), and soil dehydrogenase (S-DHA) activity. The alkaline hydrolysis diffusion method was used to determine QN [23]. QP was quantified using molybdenum-antimony spectrophotometry [24]. Cu, Fe, and Ca were determined using atomic absorption spectrophotometry [25]. A conductivity monitor was used to identity the EC [26]. OM was determined by oxidation method [27]. Chloroform fumigation extraction was used to assess MBC [28]. S-ALP was determined by colorimetric method [29]. S-SC was determined by colorimetric method using 3,5-dinitrosalicylic acid [30]. S-UE was determined by colorimetric method using indanol blue [31]. S-DHA was assessed by colorimetric method [32]. In every soil plot, measurements of the soil’s characteristics were made three times. The above experimental technology service was provided by Norminkoda Biotechnology Co., Ltd. (Wuhan, China).

### 2.3. DNA Extraction and PCR Amplification

Microbial community genomic DNA was extracted from individual pepper assemblage samples using the CretMagTM-Powered Soil DNA Kit ((CretBiotech, Suzhou, China). DNA extracts were assayed on 1% agarose gels for DNA extracts, and a NanoDrop 2000 UV-visible Spectrophotometer (Thermal Sciences, Inc., Wilmington, NC, USA) was used to determine the concentration of DNA and the purity. The bacterial 16S was amplified by an A200 PCR thermocycler (A200, LongGene, Hangzhou, China) using primers 341F (5′-CCTAYGGGRBGCASCAG-3′) and 806R (5′-GGACTACHVGGGTWTCTAAT-3′). rRNA gene with highly variable region V3-V4 [33]. The hypervariable regions of the ITS were amplified with primer pairs ITS1F(5′-CTTGGTCATTTAGAGGAAGTAA-3′) and ITS2R (5′-GCTGCGTTCTTCATCGATGC-3′) by an A200 PCR thermocycler (A200, LongGene, China) [34]. The PCR amplification procedure for the genes was as follows: initial denaturation at 94 °C for 2 min, followed by 30 cycles of denaturation at 94 °C for 30 s, annealing at 55 °C for 30 s, extension and annealing at 72 °C for 45 s, and a single extension at 72 °C for 10 min ending at 4 °C.

### 2.4. Illumina Novaseq Sequencing

Purified amplicons were combined in equimolar ratios and subjected to paired-end DNA sequencing on the Illumina MiSeq PE300 platform (Illumina, San Diego, CA, USA) using protocols established by Baiaoweifan Biotechnology Co., Ltd. (Wuhan, China). The original sequencing reads were submitted to the NCBI Sequence Read Archive (SRA) database (Accession Number: PRJNA1211196).

### 2.5. Sequencing Data Processing

The original 16S rRNA gene sequencing readings underwent demultiplexing, quality filtering using fastp version 0.20.0 [35], and merging via FLASH version 1.2.7 [36] according to the following specified criteria: (i) the 300 bp reads were trimmed whenever the mean quality score dropped below 20 within a 50 bp sliding window. Reads less than 50 bp after trimming were removed, and reads with unclear characters were excluded; (ii) only overlapping sequences exceeding 10 bp were constructed based on their overlapping regions. The highest permissible mismatch ratio in the overlap region is 0.2. Unassembled reads were eliminated; and (iii) samples were categorized based on barcodes and primers, sequence orientation was corrected, precise barcode matching was ensured, and a two-nucleotide mismatch was permitted in primer matching.

Operational taxonomic units (OTUs) were clustered at 97% similarity threshold using UPARSE version 7.1 [37]; chimeric sequences were uncovered and eliminated. The taxonomy for every OTU-representative sequence was assessed using RDP Classifier version 2.2 [38] against the 16S rRNA database with a confidence level of 0.7.

### 2.6. Statistical Analyses

Statistical analyses were performed using IBM SPSS 20.0 (IBM Corporation, New York, NY, USA) and R software (version 3.5.2). One-way analysis of variance (ANOVA) and post hoc multiple comparison tests (Tukey HSD) were used to analyze the effect of anthracnose on soil properties and enzyme activities. Wilcoxon rank sum test was used as a test of statistical significance for comparison of the three groups. All statistical tests conducted in this study were considered significant at *p* < 0.05 and extremely significant at *p* < 0.01.

## 3. Results

### 3.1. Comparative Analysis of the Physical and Chemical Properties of HR, R, and S

Sixteen features of the inter-root soils of each group were evaluated—QN, QP, QK, Ca, Mg, Zn, Fe, Cu, soil pH, OM, EC, MBC, S-SC, S-ALP, S-UE, and S-DHA activity.

These were used to ascertain the correlation among chemical characteristics, enzyme activities, and the prevalence of anthracnose in peppers. The results in the graph (Figure 1) demonstrated that substantial alterations occurred in some of the soil physicochemical features and enzyme activities in HR, R, and S. Calcium, magnesium, zinc, and iron contents in HR, R, and S were reduced in that order, and the differences were significant. In the measurement of soil enzyme activities, no substantial differences were observed, except that the S-ALP concentration in group R was markedly higher than that in groups HR and S. Additionally, the difference in inter-root soil pH of pepper across three distinct health situations was not significant (*p* > 0.05), suggesting that the presence of anthracnose in pepper did not influence the inter-root soil pH.

### 3.2. Analysis of Microbial Diversity and High-Throughput Sequencing Data in HR, R, and S

The species accumulation curves for the bacterial and fungal community sequencing results were plotted (Figure 2A,E) to assess the adequacy of the sample size in this study and to estimate community richness. The findings demonstrated that the number of newly discovered species essentially hit a plateau when the number of samples was above ten. Nine samples were adequate in this investigation to represent the species composition of the community. We drew Venn diagrams to examine common and unique species in the three samples. The bacterial community generated 3711 OTUs (Figure 2B), and the fungal community generated 2283 OTUs (Figure 2F). We calculated alpha-diversity indices using QIIME2 (qiime version v.1.8.0) and plotted alpha-diversity index box plots to reflect changes in the richness and variety of microbial communities within the samples. The Chao1 and ACE indices represent the abundance of microbial communities, and the Shannon and Simpson indices represent the variety of microbial communities. It can be seen from the figure that in the bacterial community, the Chao1 index and ACE index of HR were markedly elevated compared to S and R (Figure 2C). The Shannon diversity index of HR was markedly greater than that of S and R. However, in comparison of the Simpson indices, R had the lowest index, i.e., the highest diversity, but it was not statistically significant due to *p*-value greater than 0.05 (Figure 2D). Among the fungal communities, HR had higher abundance and diversity than H and R, but none of the results were statistically significant.

### 3.3. Cluster Analysis of Microbial Communities and Similarities and Differences in Their Structures in HR, R, and S

The multidimensional soil microbial variables were downscaled into two variables by principal coordinate analysis (PCoA) (Figure 3) of microbiological consortia from three different plots. For the bacterial community, the contribution of the first principal coordinate (PCoA1) was 30.67%, the contribution of the second principal coordinate (PCoA2) was 12.33%, and the cumulative contribution of the two was 43.00%. From Figure 2A, it is evident that the soil samples from HR and R, S areas were farther apart from each other, indicating that the disparity in bacterial community makeup between the highly resistant sample plots and the resistant and susceptible sample plots was more pronounced. Regarding the fungal community, the contribution of PCoA1 was 37.2%, that of PCoA2 was 12.7%, and the cumulative contribution of the two was 49.9%. As can be seen in Figure 3B, the soil samples from the HR area and the R, S area were farther apart from each other, indicating that the distinction in fungal community composition between the two sample sites was more pronounced. This is consistent with the bacterial results.

### 3.4. Community Composition and Abundance of Dominant Species in HR, R, and S

To examine the community structure of microorganisms in the soil across the three sample groups, we analyzed the relative abundance of soil bacteria and fungus classified by genus and phylum (Figure 4). For the bacterial community, there are 30 bacterial phyla with proportional abundance greater than 1% at the phylum standard. The predominant phyla were *Acidobacteriota* and *Proteobacteria*. As seen in the heatmap, *Acidobacteriota* showed a gradual decrease in relative abundance in HR, R, and S (Figure 4A). At the genus level, there are 30 genera of bacteria with relative abundance above 1%. The genera with higher relative abundance were *Vicinamibacteraceae*, *RB41*, and *Vicinamibacterales*. As seen in the heatmap, the proportional abundance of *Subgroup-2*, *AD3* in HR and R was greater than that in group S, but the proportional abundance of *MND1* in group S was higher than that in groups HR and R (Figure 4B). For the fungal community, there are 18 fungal phyla with relative abundance above 1% at the phylum-level standard. Among them, the predominant phyla were *Ascomycota* and *Basidiomycota*. As seen in the heat map, the relative abundance of *Ascomycota* in HR, R, and S gradually increased, and the relative abundance of *Basidiomycota* in HR, R, and S gradually decreased (Figure 4C). At the genus level, there are 30 genera of fungi with relative abundance exceeding 1%. The genus with high relative abundance were the fungi, uncultured *Botryotrichum_atrogriseum*. As can be seen from the heatmap, the relative abundance of uncultured *Agaricales* and uncultured *Trechisporates* was higher in HR than in R and S.

### 3.5. Functional Analysis of the HR, R, and S Microbial Communities

We compared the functional abundance of bacterial and fungal communities based on sequence analysis of the ITS markers of the 16S rRNA gene. PICRUSt 2 (systematic genetic Investigation of communities by Reconstruction of observed States 2) and FUNGuild (Fungi Functional Guild) V1.0 [30] online platform were utilized to forecast microbial function, thereby establishing a foundation for comprehending the microbial community and its potential relationships with host peppers. For the bacterial communities, we plotted histograms of differential function (Figure 5A). The graphs show functional genes with significantly different abundances in different groups, and the length of the bars represents the relative abundance of significantly different functional genes. As seen from the figure, the functions of bacterial chemotaxis and flagellar assembly in HR were significantly higher than those of R and S. Geraniol degradation and synthesis, the degradation of ketone bodies, and the citrate cycle were found to be in high functional abundance in S. Based on ANOVA analysis of functional gene abundance (Figure 5B), the functional abundance of aminoacyl-tRNA biosynthesis, alanine, aspartate, and glutamate metabolism were all higher in functional abundance, but none were significantly different.

FUNGuild is a database providing functional annotations for fungi, encompassing information for approximately 12,000 species. It predicts fungal functionality by delineating the structural makeup of fungal communities for annotation, and we analyzed the functional prediction results of three sample plots and performed ANOVA analysis (Figure 5C,D). The findings indicated that the functional abundance of arbuscular mycorrhizal was markedly greater in HR compared to R and S. The animal pathogen-dung saprotroph-endophyte-epiphyte-plant saprotroph-wood saprotroph had higher but not significantly different functional abundance.

## 4. Discussion

Pepper (*Capsicum* spp.) is popular for its unique flavor, and demand is increasing [39]. With the expansion of the scale of pepper production, pepper anthracnose is also gradually spreading, becoming a major problem hindering the healthy development of the pepper industry [40]. Pepper varieties, continuous cropping, sticky soil, heavy treatment and light prevention, and failure to clean up leaves with *Anthracnose germs* and residual leaves in time can cause anthracnose to occur [41,42]. Chemical control serves as the primary method for disease management. Excessive use of chemical agents not only damages the ecological environment but also causes harm to humans [43]. Biological control is frequently regarded as a more economical option relative to chemical and physical methods [44]. Microorganisms significantly contribute to plant disease resistance [45]; for example, the induction of disease resistance in goldenseal by an antagonistic bacterium of stem rot of the nematode tree [46]. Therefore, it is essential to investigate the physicochemical properties and microbial community composition of the inter-root soil of pepper to facilitate the biological control of pepper anthracnose.

### 4.1. Relationship Between Physical and Chemical Properties of Inter-Root Soil and Plant Health

Microbial activity and plant growth are significantly impacted by the physicochemical characteristics of the soil and the activities of its enzymes [34]. Quick-acting nutrients in soil mainly include soil alkaline exchangeable calcium/magnesium, effective iron, quick-acting potassium, alkaline dissolved nitrogen, etc. [47]. Lack of nutrients tends to result in the creation of plant diseases. Yang Fan et al. in their investigation into the impact of soil fast-acting nutrient content on the yield of peppers found that the number of diseased fruits of peppers in the experimental area lacking phosphorus fertilizer ratios was relatively high [48]. One study showed that calcium deficiencies can lead to blossom rot in tomatoes [49]. This aligns with the findings of the current investigation, in which calcium, magnesium, zinc, and iron contents in HR, R, and S sequentially decreased with significant differences. However, the differences in N, P, and K were not significant, which may pertain to the composition of beneficial nutrients in the soil or the nutrient absorption mechanism of the pepper plant, and the relationship between these three factors and anthracnose needs to be further studied. The activity of various soil enzymes is a crucial factor for plant growth and development [50]. Additionally, they contribute to the promotion of benign soil ecological cycles and the prevention of pest diseases [51]. S-ALP is considered to be an important driver of the microbial activation process of organophosphorus and can increase the effective phosphorus content of soil [52]. In this experiment, the content of phosphatase in the inter-root soil of disease-resistant peppers was markedly elevated compared to that in disease-susceptible areas, which aligns with findings from prior research, and we hypothesized that it was because the high content of the enzyme suppressed the development of anthracnose. There was no significant correlation between the three treatment groups of S-SC, S-UE, and S-DHA with resistant and susceptible soils, indicating that the activities of these three enzymes were not influenced by variations in vegetation susceptibility to disease. Studies have shown that alkaline phosphatase activity is negatively correlated with soil pH [53], which is in general agreement with the results of this experiment. Lacey et al. found that disease severity decreased with decreasing acidic pH [54]. Senechkin et al. found that wilt incidence was lower in high-pH soils [55]. It has also been suggested that there is no correlation between the severity of plant diseases and pH [56], and the results of the present experiments are in agreement with that result. We also determined the electrical conductivity (EC) and the organic matter content of the soil. High EC values create a reverse osmotic pressure, which dries out and browns the root system, preventing it from absorbing water and nutrients [57]. N increase in soil organic matter content improves plant growth and yield [58]. However, there was no significant difference between the two results of this experiment. The activities of most enzymes in soil are affected by soil microbiomass carbon [51], which increased with the increase of plant disease resistance in this experiment. Therefore, we hypothesized that microbial amount carbon MBC plays an important role in anthracnose suppression.

### 4.2. Diversity and Community Composition of Inter-Root Soil Microorganisms in Relation to Plant Health

Inter-root microorganisms are regarded as one of the most intricate ecosystems on Earth, encompassing a multitude of biological species and serving as a conduit between the soil and the plant [59]. The variety and makeup of soil microorganisms influence the multifunctionality of soil ecosystems and suppress plant diseases [60]. Diverse microbial community composition has been demonstrated to more effectively inhibit pathogens [59]. Zhou xin et al. found that bacterial and fungal diversity were higher in the inter-root soil of tomato with lower incidence [61]. Chen shu juan et al. discovered that the lowest microbial diversity was found in the inter-root soil of poorly growing blueberries [62]. In this experiment, disease-resistant soil had more bacteria in terms of both abundance and diversity than disease-susceptible soil; however, the diversity difference was not statistically significant. The results of abundance and diversity of fungi were the same as those of bacteria, both of which were higher in disease-resistant than in susceptible soils, but the differences were not significant. The results indicate that disease-resistant varieties have a greater ability to recruit beneficial microorganisms under susceptible soil conditions. This is consistent with previous results, but the difference was not significant. We hypothesize that it is due to the fact that these factors may not be consistent indicators [63].

Further species analysis of soil bacteria in the different treatments at the phylum taxonomic level revealed that *Acidobacteriota* and *Proteobacteria* were the dominant bacteria in the three treatments, which is consistent with the findings of earlier research [64,65,66]. *Acidobacteriota* and *Proteobacteria* can be used as indicators of soil nutrient status, with *Acidobacteriota* as nutrient-poor bacteria that can expand across a broad range of amounts of carbon sources [67]. As eutrophic bacteria, the relative abundance of *Proteobacteria* usually increases at high nitrogen levels [68]. With a variety of metabolic and genetic roles, *Acidobacteriota* are a significant component of the soil bacterial community [69], usually associated with disease suppression [70,71]. In this experiment, the proportional quantity of *Acidobacteriota* in the soil increased as the degree of disease resistance increased. We hypothesized that this was due to the inhibition of pepper anthracnose by *Acidobacteriota*. At the genus level, *Vicinamibacteraceae*, *RB41* were the predominant bacterium found in healthy plants’ inter-root soil. *Vicinamibacteraceae* belongs to *Acidobacteriota*, which was found by Yang Fajun to be closely related to plant uptake of nitrogen, phosphorus, and potassium [72]. Wang Jing et al. found that this bacterium may better promote inter-plant root uptake of nutrients, which is closely linked to the utilization of nitrogen and phosphorus nutrients [73]. Wang Honglan et al. found in the study of inter-root soil of Qiangwu that its function was related to promoting soil nutrient decomposition, conversion utilization and uptake, and improving plant resistance to stresses [74]. Synthesizing previous studies, we hypothesized that the high enrichment of *Vicinamibacteraceae* in disease-resistant soils promotes the uptake and utilization of soil nutrients, which consequently promotes plant growth and enhances plant resistance. *RB41* is a common and abundant soil bacterium that is able to utilize carbon sources and nutrients in the soil and participates in the soil carbon cycle, accounting for more than half of the carbon utilization in the soil, along with *Bradyrhizobium* and *Streptomyces* [75]. As stated by Liu Wei et al., *RB41* abundance was positively correlated with urease, which may indirectly increase soil alkaline dissolved nitrogen by affecting soil carbon cycling [76]. In this study, *RB41*, urease, and quick-acting nitrogen levels were higher in disease-resistant inter-root soils than in disease-susceptible soils, consistent with their results. It has been shown that the genus *MND1*’s abundance enhances the quality of the soil. They are crucial for the solubilization of inorganic phosphorus, as well as the mineralization of organic phosphorus [77]. Furthermore, they are engaged in nitrogen cycling activities, such as ammonia oxidation and nitrogen fixing [78]. In this study, however, the amount of *MND1* in sensitive soil was higher than that in healthy soil. We hypothesize that this is due to the influence of the complex microbial ecological network structure, about which the specific mechanism of influence remains to be studied.

The makeup of fungal communities is simpler than that of bacterial communities. *Ascomycota* and *Basidiomycota* appeared as the main endophytic fungal dominant groups, which agrees with the findings of Yao Suhang et al. [79]. *Ascomycota* has high species diversity and evolutionary rate [80] and dominates soil fungal communities globally [81]. It is not only highly adaptable to arid environments but also plays an important function in decomposing soil organic matter [82]. *Ascomycota* are essential for the degradation of plant apoplasts with high lignin content [83]. Ectomycorrhizal fungi (ECMF), which are also included in *Ascomycota* and *Basidiomycota*, form a mutually beneficial symbiotic relationship with plant roots and have a significant part to play in the nitrogen and phosphorus cycle [84]. It is therefore a key fungal taxon for pepper resistance to anthracnose. Compared to disease-susceptible soils, disease-resistant inter-root soils had a substantially higher relative abundance of uncultured *Basidiomycota* and *Trechisporates*. *Basidiomycota* play an important role in the breakdown of cellulose and lignin and are an integral part of forest ecosystems [85]. Uncultured *Trechisporates* is an unclassified order under the *Basidiomycota* whose functional studies have not been clarified and need to be further explored.

### 4.3. Relationship Between Inter-Root Soil Microbial Function and Plant Health

Much research has been carried out to show that the functional composition of microorganisms is intimately related to the environment [86,87]. We examined the functional abundance of bacterial and fungal communities using sequencing analysis of ITS markers of 16S rRNA genes. PICRUST 2 and FUNGuild predictions showed that soil bacterial and fungal communities were rich in functional diversity. Ansamycins are a class of macrocyclic lactam antibiotics mainly produced by microorganisms with complex structures, almost all of which have significant physiological activities [88,89]. Many of its derivatives have been used in clinical treatments [90,91]. The biosynthesis function of biosynthesis of ansamycins is in high abundance in both disease-resistant and disease-sensitive soil areas. We hypothesize that the inter-root soil of pepper may produce certain metabolites or enrich a certain microflora closely associated with the synthesis of such compounds. Furthermore, the functional abundance of biosynthesis of vancomycin group antibiotics, valine, leucine, and isoleucine biosynthesis were also high. Vancomycin is a glycopeptide antibiotic, which can prevent the bacterial cell wall from synthesizing and from producing polypeptides and phospholipids so as to kill bacteria and avoid bacterial infection [92]. Valine, leucine, and isoleucine as branched-chain amino acids (BCAAs) are biosynthetically important in plant disease resistance defense. Naoki et al. found that exogenous proteogenic amino acids can confer plant resistance, such as isoleucine and leucine, that trigger resistance to rice blast. All of these functions enhance plant disease resistance.

Functional prediction of FUNGuild revealed that the inter-root soil of susceptible plants with animal pathogen-dung saprotroph-endophyte-epiphyte-plant saprotroph-wood saprotroph had maximum functional abundance. These plant diseases may have a direct bearing on this outcome. In healthy plant inter-root soil, arbuscular mycorrhizal had a much higher functional abundance than in R and S. Arbuscular mycorrhizal are the most widely distributed soil symbiotic fungi on earth; they can form symbiotic relationships with more than 80% of terrestrial plants, absorb carbohydrates produced by plant photosynthesis through the symbiotic relationship, and transport the photosynthesized products downward to convert them into difficult-to-biodegrade organic matter, and their clumped structures near the root system are considered to be an important place for plant nutrient exchange [93]. In addition, arbuscular mycorrhizal accelerate soil organic matter decomposition, and the associated soil proteins secreted after apoptosis can be released into the soil as a carbon source [94]. It can also alleviate the growth inhibition of plants under stressful environments to varying degrees and improve plant resistance and adaptability [95]. Weili et al. found that arbuscular mycorrhizal reduced anthracnose in tea seedlings [96]. Yang Yalin et al. found that there was a decreasing trend in the abundance and diversity index of AMF spores in disease-susceptible oil tea soil [97]. Based on previous studies, we hypothesized that it might be due to the fact that the arbuscular mycorrhizal competed with the pathogen for ecological niches at the inter-root level, reducing the chances of infestation by the pathogen.

Of course, interpreting these variations as merely soil–microbe interactions is insufficient; the direct effects of these microbial and chemical characteristics on plant health require more study. We combined correlation analyses to develop a correlation mechanism map (Figure 6) that reveals possible pathways between environmental factors, inter-root bacteria and fungi, and pepper resistance and susceptibility to anthracnose. Whether specific microbial communities and soil factors in healthy pepper inter-root soils have the ability to suppress disease and the optimal microbial ecological network structure remain to be further investigated.

## 5. Conclusions

Despite using the same growth conditions for peppers, there were significant differences observed in soil physical and chemical properties and microbial communities under the three different health conditions. MBC increased as plant disease resistance increased, and soil MBC levels may inhibit anthracnose development. The relative abundance of *Vicinamibacteraceae* and *RB41* is higher in healthy soils. This promotes the decomposition, translational utilization, and uptake of nutrients such as nitrogen, phosphorus, and potassium. At the fungal taxonomic level, *Ascomycota* and *Basidiomycota*, as the main dominant groups of endophytic fungi, play an important role in the decomposition of soil organic matter, the degradation of lignin, and nitrogen and phosphorus cycling. Differences in function are mainly manifested in apoptosis and arbuscular mycorrhizal fungi.

In conclusion, the correlation analysis and functional prediction of key environmental factors, microbiota, and pepper health indices in the inter-root soil of peppers deepened the understanding of the inhibitory effects of soil microorganisms on *Capsicum* anthracnose and provided a reference for the management of peppers infections. In addition, we predicted interactions between anthracnose, soil factors, and inter-root microbial communities, providing a basis for future prevention of anthracnose through microecological regulation.

## Figures and Tables

**Figure 1 microorganisms-13-00661-f001:**
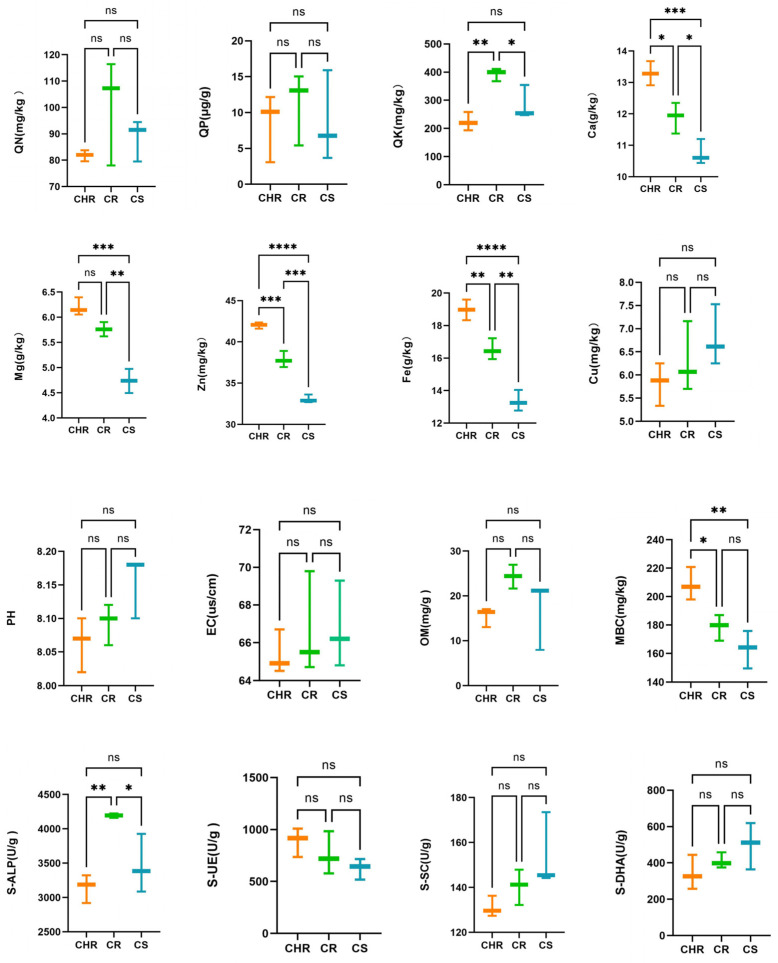
Effect of resistance and susceptibility to pepper anthracnose on chemical characteristics of inter-root soil and enzymatic activities. The symbol ‘*’ denotes significant differences, the symbol ‘ns’ indicates no significant difference. (*p* < 0.05) across the three groups as determined by one-way ANOVA and HSD test. * *p* < 0.05, ** *p* < 0.01, *** *p* < 0.001, **** *p* < 0.0001.

**Figure 2 microorganisms-13-00661-f002:**
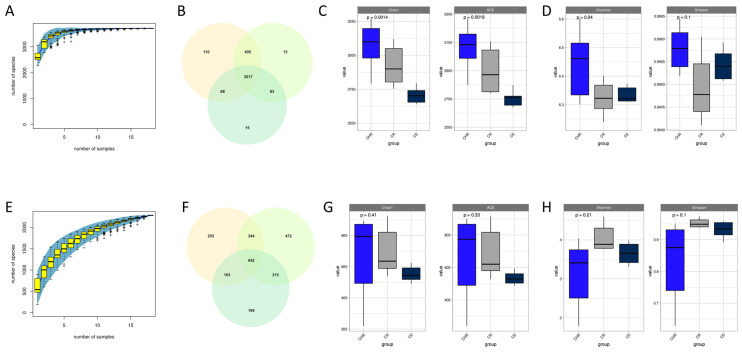
α-diversity of inter-root microbial communities. Species accumulation curves for bacteria (**A**) and fungi (**E**). Venn diagrams were constructed to illustrate the shared and unique bacterial (**B**) and fungal (**F**) taxa identified across the three groups. Box plots illustrate the variation in the Chao1 index, ACE index, and Shannon and Simpson indices for bacteria (panels (**C**,**D**)) and fungi (panels (**G**,**H**)) across the three groups.

**Figure 3 microorganisms-13-00661-f003:**
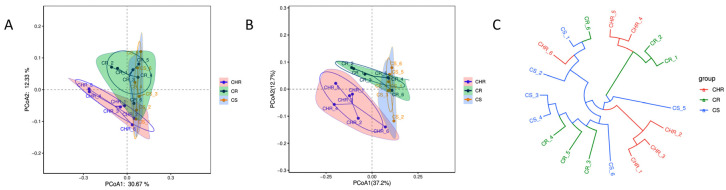
Plot of PCoA analysis of bacteria (**A**) and fungi (**B**). The horizontal and vertical coordinates represent the first and second principal components and their contribution to the sample difference, respectively. The branch lengths of the Bray–Curtis multi-sample clustering tree (**C**) for bacteria indicate the distance between samples; greater similarity across samples increases the likelihood of their grouping together.

**Figure 4 microorganisms-13-00661-f004:**
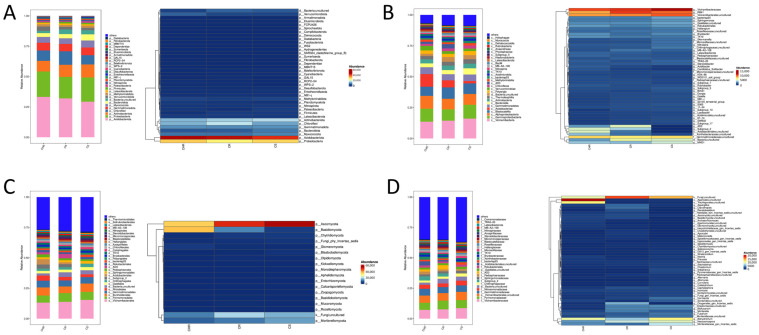
Composition of microbial communities at the phylum and genus level (top 30 by relative abundance). Bacteria at the phylum (**A**) and genus (**B**) classifications. Fungi at the phylum (**C**) and genus (**D**) level.

**Figure 5 microorganisms-13-00661-f005:**
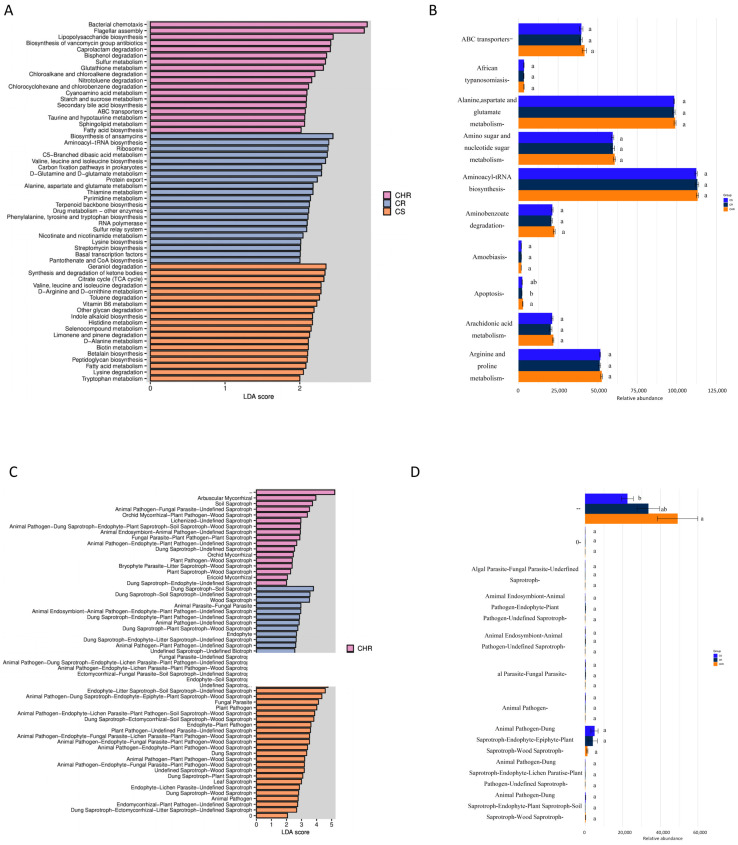
Prediction and analysis of metabolic function based on Picrust II and FUNGuild for bacterial communities (**A**,**B**) and fungal communities (**C**,**D**) of HR, R, and S. The effect values of linear discriminant analysis (LEfSe) were greater than 2.0. The linear discriminant analysis effect values (LEfSe) for bacterial (**A**) and fungal (**C**) communities exceeded 2.0 when the *p*-value was below 0.05. The histograms illustrate the linear discriminant analysis scores of the microbial communities. Communities that met the LDA significance threshold greater than 2.0 are shown. ANOVA analysis derived from the prevalence of functional genes in bacteria (**B**) and fungi (**D**). The horizontal coordinates indicate the relative abundance of a species in different subgroups, the vertical coordinates indicate the differential species, and different colors indicate different groups. Different letters indicate the significance of differences.

**Figure 6 microorganisms-13-00661-f006:**
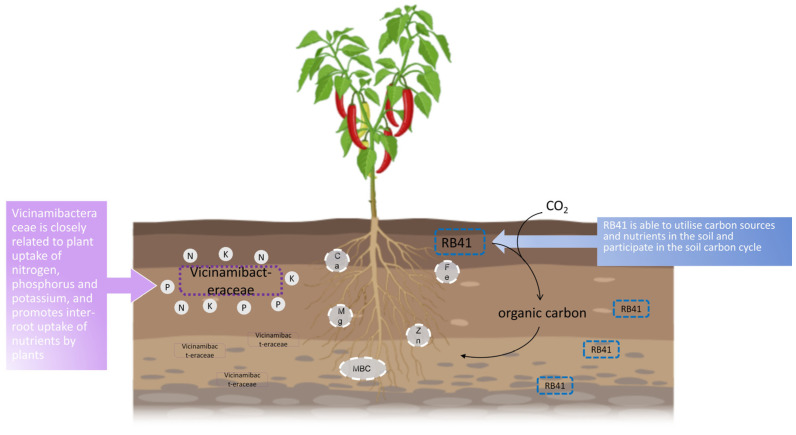
Diagram of possible mechanisms of inhibition of pepper anthracnose by inter-root soil bacteria and fungi in combination with soil factors. *Vicinamibacteraceae* is closely related to plant uptake of nitrogen, phosphorus, and potassium and promotes inter-root uptake of nutrients by plants. *RB41* is able to utilize carbon sources and nutrients in the soil and participate in the soil carbon cycle.

## Data Availability

All raw sequence data have been made available in the NCBI Sequence Read Archive (SRA) database under the bioproject accession number PRJNA1211196.

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
