# Peer review of "Influence of Soil Physicochemical Properties and Inter-Root Microbial Communities on the Inhibition of Anthracnose in Peppers"

_microorganisms, 2025, doi:10.3390/microorganisms13030661_

Round 1
Reviewer 1 Report
Comments and Suggestions for Authors
The manuscript studies important topic of microbial-crop diseases which is still worth of study in order to better know such problems and to improve crops quality and quantity. As as soil, especially rhizosphere, is a amazing niche for numerous microbes often affecting plant growth, any studies focusing of this part of environment are one of the most wanted. Authors clearly identified root-related microbes are responsible for crop diseases and focused to study plant-microbial interaction. This is clearly highlighted in Introduction ending with a clear aim.
This makes this paper falling into the scope of Microorganisms MDPI Journal. The text is well organized in a way of scientific, research paper with all required parts included. It is written with good English and is scientifically sound.
Title and keywords are well done. The abstract is quite informative providing a summary about the work done. There are abbreviations of treatments (HR, R, and S) which should be explained as they appear for the first time. Whats more, material and methods present different codes, i.e. NHR, NR and NS. Please clarify this.
The sentence in line 35-36 "Differences in function are mainly manifested in apoptosis, Arbuscular Mycorrhizal." is not clear to me. What is about 'Arbuscular Mycorrhizal' in this sentence? Shouldn't it be Arbuscular Mycorrhizal Fungi? But still I don't understand this sentence.
When providing Latin names of organisms on the species and genus level it is common to italize them - for instance, in line 47 - Colletotrichum should be italzed. Please check the whole manuscript for similar issues.
L80 - please write 'actinomycetes' with capital letter (Actinomycetes)
Material and methods
When checking the whole manuscript I noticed that the soil treatments have different codes - different in abstract, text and on figures (e.g. NHR, HR, CHR = the same treatment). Please clarify this issue and unify them.
There are not so much information about soil sampling:
- the number of soil samples (sampling points) collected and their quantities (masses)
- what was the number of replicated for each treatment?
- did you sterilize soil sampler prior soil collection and between each spot?
You didn't mention anything about pepper varietes, also no information about spots selection for sampling is provided. Did you selected both healthy and infected plants? How did you distinguish between high resistant, resistant and susceptible plants?
There is no information about growing conditions of peppers.
In conclusion, there are many gaps in experimental design and sample selection which are crucial steps in any research studies.
L118 - you mentioned 16 characteristics determined and listed them. When I counted them, only 15 are listed. Next you just mentioned some methods, more precisely only their principles. When describing experiments the methods used should be presented in the way allowing it repetition by other scientists. In addition, no models and manufacturers of analytical equipment used mentioned. This makes difficult to judge if the proper methods were used and to know the level of accuracy and precision of analytical equipment used. This must be clarified.
L137 - please add space between "China)." and "DNA extracts..."
L144 - please remove ". ')"
There are missing citation about data processing - statistics and bioinformatics. Authors used some statistical tests but no software is mentioned. The some text about statistical methods is also required. The same applies for bioinformatics. Have you checked the requirements for parametric test such as ANOVA? Why did you use less-conservative LSD, HSD is much robust?
Results are supported by nice graphs full of details and explained in their captions. Some of them, e.g. Fig 2, 3 have very small and blurry font and are hard to read. Other figures such as Fig. 4, 5B,D have so small font that cannot be read. Please improve them.
Obtained data are quite well described and supported with statistical analysis, sometimes it would be worth add more details, some numbers to make the results more sound. Discussion is ok to me leading to nice home-message (conclusions are part of discussion) supported by Fig. 6
Author Response
We would like to thank the reviewer for careful review of our manuscript and providing us with his constructive comments and suggestions to improve the quality of our manuscript. The following responses have been prepared pertinent to all of the reviewer’s comments in a point-by-point fashion.
Point-by-point response to Comments and Suggestions for Authors:
Comments 1: Title and keywords are well done. The abstract is quite informative providing a summary about the work done. There are abbreviations of treatments (HR, R, and S) which should be explained as they appear for the first time. Whats more, material and methods present different codes, i.e. NHR, NR and NS. Please clarify this.
Response 1: Thank you for kindly reminding us. We have explained the abbreviations that appear for the first time. (Line108-109) In addition, we harmonized the different codes for material and method presentation.
Comments 2: The sentence in line 35-36 "Differences in function are mainly manifested in apoptosis, Arbuscular Mycorrhizal." is not clear to me. What is about 'Arbuscular Mycorrhizal' in this sentence? Shouldn't it be Arbuscular Mycorrhizal Fungi? But still I don't understand this sentence.
Response 2: We feel great thanks for your professional review. The Arbuscular Mycorrhizal here refers to mycelial network function, and it was our poor wording that made the meaning here difficult to understand, and we have made changes. (Line34)
Comments 3: When providing Latin names of organisms on the species and genus level it is common to italize them - for instance, in line 47 - Colletotrichum should be italzed. Please check the whole manuscript for similar issues.
Response 3: We sincerely thank the reviewer for careful reading. We've modified it to italics and checked the whole manuscript to modify similar issues. (Line44, Line539)
Comments 4: L80 - please write 'actinomycetes' with capital letter (Actinomycetes)
Response 4: Thanks for pointing this out. We have changed it as you requested. (Line76)
Material and methods
Comments 5: When checking the whole manuscript I noticed that the soil treatments have different codes - different in abstract, text and on figures (e.g. NHR, HR, CHR = the same treatment). Please clarify this issue and unify them.
Response 5: We sincerely thank the reviewer for careful reading. We apologize that due to an oversight on our part, the same processing was represented in different code. We have harmonized them. C
Comments 6: There are not so much information about soil sampling:
- the number of soil samples (sampling points) collected and their quantities (masses)
- what was the number of replicated for each treatment?
- did you sterilize soil sampler prior soil collection and between each spot?
You didn't mention anything about pepper varietes, also no information about spots selection for sampling is provided. Did you selected both healthy and infected plants? How did you distinguish between high resistant, resistant and susceptible plants?
There is no information about growing conditions of peppers.
In conclusion, there are many gaps in experimental design and sample selection which are crucial steps in any research studies.
Response 6: We thank the reviewer for these important questions, and we have responded to them individually
- The number of soil samples collected was 15 (five samples from each of the three groups HR, R, and S were taken by S-type five-point random sampling method). The sampling site was an experimental base for the production and cultivation of peppers at the Henan Provincial Academy of Agricultural Sciences, China.
-Three replications were performed for each treatment.
-Soil samplers were sterilized prior to soil collection and between each point to ensure that there was no interaction between the front and back samples.
The variety of pepper was “Sanying-1 pepper”, and the selection of sampling sites was based on the different incidence rates of peppers. Areas with 0% incidence are high resistant, 0-5% incidence is resistant, and more than 50% incidence is susceptible.
The “Sanying-1” peppers used in this study were grown under normal outdoor conditions at the site without any special treatments or artificially controlled environmental settings. These pepper plants were completely exposed to natural light, temperature, humidity and precipitation conditions to mimic the conventional agricultural production environment.
We have added information on experimental design and sample selection based on your suggestions, and we hope these additions will fulfill your requirements. Thank you again for your professional advice! (Line104-106, Line109-110)
Comments 7: L118 - you mentioned 16 characteristics determined and listed them. When I counted them, only 15 are listed. Next you just mentioned some methods, more precisely only their principles. When describing experiments the methods used should be presented in the way allowing it repetition by other scientists. In addition, no models and manufacturers of analytical equipment used mentioned. This makes difficult to judge if the proper methods were used and to know the level of accuracy and precision of analytical equipment used. This must be clarified.
Response 7: We thank the reviewer for bringing this to our attention. The 16 characteristics included here are QN, QP, QK, Ca, Mg, Zn, Fe, Cu, PH, OM, EC, MBC, S-SC, S-ALP, S-UE, S-DHA. Calcium/magnesium content (Ca/Mg) in the manuscript is two metrics. We apologize for any misinterpretation due to the lack of clarity in our presentation. In addition, we have added references to the methods used for the determination of each indicator in the manuscript so that they can be reused by other scientists, and we have indicated the companies that provided the technical services mentioned above. (Line125-134)
Comments 8: L137 - please add space between "China)." and "DNA extracts..."
Response 8: We sincerely thank the reviewer for careful reading. We have added space between "China)." and "DNA extracts..." (Line138)
Comments 9: L144 - please remove ". ')"
Response 9: Thanks to the reviewer for pointing this out. We have removed ". ')" (Line145)
Comments 10: There are missing citation about data processing - statistics and bioinformatics. Authors used some statistical tests but no software is mentioned. The some text about statistical methods is also required. The same applies for bioinformatics. Have you checked the requirements for parametric test such as ANOVA? Why did you use less-conservative LSD, HSD is much robust?
Response 10: We thank the reviewer for these thought-provoking suggestions. We have added citations on data processing - statistics and bioinformatics. (Line158 Line168, Line170)
The software and methods used for the statistical tests have also been shown in the manuscript. (Line171-178)
Regarding the requirement for parametric tests, we did use HSD. it was our confusion between the two methods of analysis that led to the error in the writing in the manuscript, and we thank you for raising this professional issue.
Comments 11: Results are supported by nice graphs full of details and explained in their captions. Some of them, e.g. Fig 2, 3 have very small and blurry font and are hard to read. Other figures such as Fig. 4, 5B,D have so small font that cannot be read. Please improve them.
Response 11: Thanks to the reviewer for the kind reminder. We have increased the clarity of the diagrams and the font size to enable them to be read clearly. However, because the diagrams contain more important information, the font size could not be further increased, and we suggest readers to zoom in appropriately when viewing the diagrams in order to better read the details. We understand that the readability of graphs is crucial for readers to understand the results of the study, and we will pay more attention to the design and layout of the graphs in the preparation of subsequent manuscripts to ensure that all the graphs are optimized in terms of clarity and presentation of information.
Comments 12: Obtained data are quite well described and supported with statistical analysis, sometimes it would be worth add more details, some numbers to make the results more sound. Discussion is ok to me leading to nice home-message (conclusions are part of discussion) supported by Fig. 6
Response 12: Thank you to the reviewer for his positive comments and professional advice. We fully agree with your suggestion that “sometimes more details and figures need to be added to make the results more reasonable”. In the subsequent revisions, we will also pay more attention to the detailed presentation of data and the transparency of statistical analysis. In addition, you pointed out that “the conclusion is part of the discussion”, which made us realize that we might not be able to clearly distinguish the two parts of the discussion and conclusion in the manuscript, resulting in a lack of clarity in the structure. Based on your suggestion, we have added a conclusion section to clearly delineate between the discussion and conclusion sections. (Line495-513)
We appreciate for reviewer's warm work, and hope the correction will meet with approval. Once again, thank you again for your positive comments and valuable suggestions to improve the quality of the manuscript.

Reviewer 2 Report
Comments and Suggestions for Authors
In the article presented the study on phsico-chemical interactions between root microbial communities on removal of anthrsaconose in peppers. The anthraconose disease causes serious damage in field production of vegetables, flowers, fruit crops causing big economic losses on the farms. The aim of the study was to obtain friendly ecological control of mention above problem. Literature shows that relatively not much research was conducted on this field, so authors made quite a progress concerning microbial control of pepper inter-roots composition.
Please add Section Conclusions with description of the most important findings of the study.
Please correct Figure 5 and Figure 6 to make them more readible.
Author Response
We would like to thank the reviewer for scrutinizing our manuscript and providing us with constructive comments and suggestions to improve the quality of our manuscript. We have revised our manuscript in accordance with your suggestions.
Point-by-point response to Comments and Suggestions for Authors:
Comments 1: Please add Section Conclusions with description of the most important findings of the study
Response 1: We thank the reviewer for the positive feedback and constructive points. We have added a “Conclusions” section that describes the most important findings of the study. (Line495-513)
Comments 2: Please correct Figure 5 and Figure 6 to make them more readible.
Response 2: Thank you for kindly reminding us. We have resized the fonts so that the images can be read more clearly. In addition, we have added notes and other information to make the graphics easier to understand. (Line295, Line488)
We appreciate for reviewer's warm work, and hope the correction will meet with approval. Once again, thank you very much for your comments and suggestions.

Reviewer 3 Report
Comments and Suggestions for Authors
Dear authors,
Your study is valuable and have significant new findings to be of an interest for readers.
Overall, the manuscript is well written and structured but still needs some corrections aiming at improve its scientific soundness.
My recommendations are as follows:
1) Abstract: It is too long. Please, check the Instructions for author and reduce it up to 200 words.
2) Key words: They should be three to ten, to correspond to the topic but not no repeat the title
3) In the whole text:
Please, turn the Latin names of species in italic.
Add the whole binomial Latin name of pepper and the Linneus name at the begging of the Introduction alonf with its common name. In the text use only one of them - scientific name or common name, avoid mixing them.
4) Line 68: Primary zone?
5) Lines 74-75: It is not clear - "It is considered to be the most representative microorganism that most closely affects plants[13]."
6) Lines 80-81: It is not clear - "For example, actinomycetes can markedly impede the proliferation of Botrytis cinerea and suppress Botrytis cinerea[16]."
7) Lines 97-104: The aim of the study should be more precisely formulated and highlighted. The research hypothesis also needs to be strenghtened.
8) Line 100: What means "we sampled soils with different levels of incidence"? How did you assessed this? Are you sure that soils corresponded to these different levels described?
9) Line 111: Here, the samples are named as NHR, NR and NS but in the text below the names are without N (for example, see Line 119, 181, etc.)
10) Lines 112-113: Please, add a reference for this sampling method.
11) Lines 172-177 repeat the text on lines 120-124.
12) All figures in the Results section are too small. By my opinion, soem of them should be enlarged.
13) The Conclusion section is missing and shoul be added
14) References: Please, check the Instructions for authors and correct the citing style.
10) Lines 119-133: Please, add some refernces for the methods used.
11) Lines 135-148: Please, add some references here too.
Author Response
We feel great thanks for your professional review work on our article. As you are concerned, there are several problems that need to be addressed. According to your nice suggestions, we have made extensive corrections to our manuscript, the detailed corrections are listed below. In addition, we have resubmitted a new manuscript in the revised state, with the revisions highlighted in red. If there are any incorrect answers or questions in the manuscript, please do not hesitate to let us know.
Point-by-point response to Comments and Suggestions for Authors:
Comments 1: Abstract: It is too long. Please, check the Instructions for author and reduce it up to 200 words.
Response 1: Thank you for kindly reminding us. We have streamlined the abstract. (Line15-34)
Comments 2: Key words: They should be three to ten, to correspond to the topic but not no repeat the title.
Response 2: Thank you for your professional suggestion. We have changed the keywords as requested. (Line35-36)
Comments 3: In the whole text:
Please, turn the Latin names of species in italic.
Add the whole binomial Latin name of pepper and the Linneus name at the begging of the Introduction alonf with its common name. In the text use only one of them - scientific name or common name, avoid mixing them.
Response 3: We thank the reviewer for bringing this to our attention. We have checked the entire manuscript to italicize the Latin names of species. (Line39, Line44, Line76, Line78, Line254, Line257-259, Line262-263, Line266-267, Line387, Line389-390 et al.)
We have added the whole binomial Latin name and Linneus name of pepper at the beginning of the “Introduction”. (Line39-40)
In addition, to avoid mix-ups, only one of these - the scientific or common name - will be used in the body of the text, as you requested. (Line136, Line333)
Comments 4: Line 68: Primary zone?
Response 4: Mainly in the following areas:
- Agricultural field: biological insecticides have the advantages of environmental protection, safety, easy degradation, no residue, etc., in line with the development requirements of green agriculture
- Forestry field: biopesticides are used to control forest pests and protect forest resources. Its natural ingredients and low environmental impact make it an ideal choice for forestry pest management.
- Technological innovation: the progress of genetic engineering and bioinformatics promotes the discovery of new bioactive substances, which improves the selectivity and effectiveness of insecticides.
Comments 5: Lines 74-75: It is not clear - "It is considered to be the most representative microorganism that most closely affects plants[13]."
Response 5: We apologize that this sentence does not convey the meaning clearly, and we have made changes to make the sentence more concise and easier to understand. (Line70)
Comments 6: Lines 80-81: It is not clear - "For example, actinomycetes can markedly impede the proliferation of Botrytis cinerea and suppress Botrytis cinerea[16]."
Response 6: We apologize that the previous formulation was not clear enough to accurately convey our intent. We have revised the relevant content. (Line76-77)
Comments 7: Lines 97-104: The aim of the study should be more precisely formulated and highlighted. The research hypothesis also needs to be strenghtened.
Response 7: We appreciate your professional comments on our articles, which make a big difference in the quality of our articles. We have re-edited the paragraph.(Line93-101)
Comments 8: Line 100: What means "we sampled soils with different levels of incidence"? How did you assessed this? Are you sure that soils corresponded to these different levels described?
Response 8: Thanks to the reviewer for his careful reading and specialized questions. The selection of sampling sites was based on differences in the incidence of peppers. Areas with 0% incidence were resistant, 0-5% were resistant, and more than 50% were susceptible.
Comments 9: Line 111: Here, the samples are named as NHR, NR and NS but in the text below the names are without N (for example, see Line 119, 181, etc.)
Response 9: We sincerely thank the reviewer for raising this issue. We apologize that due to an oversight on our part, the same processing was represented in different code. We have harmonized them. (Line108-109)
Comments 10: Lines 112-113: Please, add a reference for this sampling method.
Response 10: Thanks to the reviewer's expert advice, we have added reference for this sampling method. (Line111)
Comments 11: Lines 172-177 repeat the text on lines 120-124.
Response 11: We thank the reviewer for reminding us of this. We have modified the repetition to make it more concise and understandable. (Line181-182)
Comments 12: All figures in the Results section are too small. By my opinion, soem of them should be enlarged.
Response 12: Thanks to the reviewer for the kind reminder. We have increased the clarity of the diagrams and the font size to enable them to be read clearly. However, because the diagrams contain more important information, the font size could not be further increased, and we suggest readers to zoom in appropriately when viewing the diagrams in order to better read the details. We understand that the readability of graphs is crucial for readers to understand the results of the study, and we will pay more attention to the design and layout of the graphs in the preparation of subsequent manuscripts to ensure that all the graphs are optimized in terms of clarity and presentation of information.
Comments 13: The Conclusion section is missing and shoul be added.
Response 13: We thank the reviewer for their constructive comments. We have added a “Conclusion” section. (Line 495-513)
Comments 14: References: Please, check the Instructions for authors and correct the citing style.
Response 14: Thanks to the reviewer's reminder, we have corrected the citation as per the “Instructions for Authors”.
Comments 15: Lines 119-133: Please, add some refernces for the methods used.
Response 15: Thanks to the professional advice of the reviewer, we have added some references to the methods used. (Line125-132)
Comments 16: Lines 135-148: Please, add some references here too.
Response 16: Thank you to the reviewer for their expert advice. As you requested, we have also added some references here. (Line143-146)
We appreciate for reviewer's warm work, and hope the correction will meet with approval. Once again, thank you again for your positive comments and valuable suggestions to improve the quality of the manuscript.

Round 2
Reviewer 1 Report
Comments and Suggestions for Authors
Dear Authors,
you have improved your manuscript and now I am satisfied.
Author Response
Response to Reviewer 1 Comments:
The quality of our manuscripts cannot be improved without your professional review, and we would like to thank you for your constructive comments and valuable suggestions!
Reviewer 3 Report
Comments and Suggestions for Authors
Dear authors,
The revised version of your manuscript was significantly improved but still needssome corections to be made.
I would like to focus on the following:
1) Lines 69-70, 102, 343, 568, 577, etc.: The Latin names of species should be in italic.
2) Lines 96-97: This sentence is not clear.
3) Lines 120-121: This sentence needs an edition.
4) Lines 122-128: By my opinion, this paragraph seems more to a syntesys of the study than to its aim description. I would like to propose something like the following: The aim of the present study was to: i) analyze the changes of environmental factors and microbiota in the inter-root soil of pepper in three different health states; ii) predict the interactions among anthracnose,
soil factors and inter-root microbial communities. Our research hypothesys was to assess the potential as a basis of preventing anthracnose in the future through microecological precision modulation. You may not to agree with my suggestions, of course.
5) Reference section needs your attention. The year of the publication should be bolded. Journal name and volume should be italicized, etc.
6) References [1], [3], [4], [6], [10] and many others - the page numbers are not fully described
7) References [14], [18], [20] - Only the first letter shoule be capitalized
Author Response
Response to Reviewer 3 Comments:
We would like to thank the reviewer for careful review of our manuscript and providing us with his constructive comments and suggestions to improve the quality of our manuscript. The following responses have been prepared pertinent to all of the reviewer’s comments in a point-by-point fashion.
Point-by-point response to Comments and Suggestions for Authors:
Comments 1: Lines 69-70, 102, 343, 568, 577, etc.: The Latin names of species should be in italic.
Response 1: We sincerely thank the reviewer for careful reading. We have checked the entire manuscript and changed the Latin names of species to italics. (Line40, Line84, Line92, Line310, Line315, Line534, Line541, Line551, Line619, Line624)
Comments 2: Lines 96-97: This sentence is not clear.
Response 2: Thank you for kindly reminding us. We have made changes to this sentence. (Line67-70)
Comments 3: Lines 120-121: This sentence needs an edition.
Response 3: Thanks to the reviewer for raising this point, and we have made changes to this sentence. (Line93-96)
Comments 4: Lines 122-128: By my opinion, this paragraph seems more to a syntesys of the study than to its aim description. I would like to propose something like the following: The aim of the present study was to: i) analyze the changes of environmental factors and microbiota in the inter-root soil of pepper in three different health states; ii) predict the interactions among anthracnose, soil factors and inter-root microbial communities. Our research hypothesys was to assess the potential as a basis of preventing anthracnose in the future through microecological precision modulation. You may not to agree with my suggestions, of course.
Response 4: We thank the reviewer for his expert advice, which made the purpose and significance of our study clearer. We have revised the manuscript according to your comments. (Line96-101)
Comments 5: Reference section needs your attention. The year of the publication should be bolded. Journal name and volume should be italicized, etc.
Response 5: Thanks to the reviewer for the kind reminder. We have corrected the formatting of the references as required.
Comments 6: References [1], [3], [4], [6], [10] and many others - the page numbers are not fully described
Response 6: We sincerely thank the reviewer for careful reading. We've filled in the references with incomplete page numbers (Line535, Line539, Line541, Line545, Line553, Line562, Line614, Line659, Line689, Line698, Line738)
Comments 7: References [14], [18], [20] - Only the first letter shoule be capitalized
Response 7: Thank you for kindly reminding us.We have made the changes you requested. (Line560-562, Line569-570, Line573)
We thank the reviewer for his careful scrutiny and professional advice and hope that the revisions will be approved. Once again, we appreciate your positive comments to improve the quality of the manuscript.
